# Visceral Leishmaniasis in pregnancy and vertical transmission: A systematic literature review on the therapeutic orphans

**Prabin Dahal**[1,2]*, **Sauman Singh-Phulgenda**[1,2], **Brittany J. Maguire**[1,2], **Eli Harriss**[3], **Koert Ritmeijer**[4], **Fabiana Alves**[5], **Philippe J. Guerin**[1,2], **Piero L. Olliaro**[2]

1 Infectious Diseases Data Observatory (IDDO), Oxford, United Kingdom, 2 Centre for Tropical Medicine and Global Health, Nuffield Department of Medicine, University of Oxford, Oxford, United Kingdom, 3 The Knowledge Centre, Bodleian Health Care Libraries, University of Oxford, Oxford, United Kingdom, 4 Médecins Sans Frontières, Amsterdam, Netherlands, 5 Drugs for Neglected Diseases initiative, Geneva, Switzerland

* prabin.dahal@iddo.org

## Abstract

### Background

Reports on the occurrence and outcome of Visceral Leishmaniasis (VL) in pregnant women is rare in published literature. The occurrence of VL in pregnancy is not systematically captured and cases are rarely followed-up to detect consequences of infection and treatment on the pregnant women and foetus.

### Methods

A review of all published literature was undertaken to identify cases of VL infections among pregnant women by searching the following database: Ovid MEDLINE; Ovid Embase; Cochrane Database of Systematic Reviews; Cochrane Central Register of Controlled Trials; World Health Organization Global Index Medicus: LILACS (Americas); IMSEAR (South-East Asia); IMEMR (Eastern Mediterranean); WPRIM (Western Pacific); ClinicalTrials.gov; and the WHO International Clinical Trials Registry Platform. Selection criteria included any clinical reports describing the disease in pregnancy or vertical transmission of the disease in humans. Articles meeting pre-specified inclusion criteria and non-primary research articles such as textbook, chapters, letters, retrospective case description, or reports of accidental inclusion in trials were also considered.

### Results

The systematic literature search identified 272 unique articles of which 54 records were included in this review; a further 18 records were identified from additional search of the references of the included studies or from personal communication leading to a total of 72 records (71 case reports/case series; 1 retrospective cohort study; 1926–2020) describing 451 cases of VL in pregnant women. The disease was detected during pregnancy in 398 (88.2%), retrospectively confirmed after giving birth in 52 (11.5%), and the time of

**Data Availability Statement:** The database(s) supporting the conclusions of this article are

available within the tables and figures presented within the manuscript along with the supplemental files (S1 Data, S2 Data).

**Funding:** The review was funded by a biomedical resource grant from Wellcome to the Infectious Diseases Data Observatory (Recipient: PJG; ref: 208378/Z/17/Z). The funders had no role in the design and analysis of the research or the decision to publish the work.

**Competing interests:** The authors have declared that no competing interests exist.

identification was not clear in 1 (0.2%). Of the 398 pregnant women whose infection was identified during pregnancy, 346 (86.9%) received a treatment, 3 (0.8%) were untreated, and the treatment status was not clear in the remaining 49 (12.3%). Of 346 pregnant women, Liposomal amphotericin B (L-AmB) was administered in 202 (58.4%) and pentavalent antimony (PA) in 93 (26.9%). Outcomes were reported in 176 pregnant women treated with L-AmB with 4 (2.3%) reports of maternal deaths, 5 (2.8%) miscarriages, and 2 (1.1%) foetal death/stillbirth. For PA, outcomes were reported in 88 of whom 4 (4.5%) died, 24 (27.3%) had spontaneous abortion, 2 (2.3%) had miscarriages. A total of 26 cases of confirmed, probable or suspected cases of vertical transmission were identified with a median detection time of 6 months (range: 0–18 months).

## Conclusions

Outcomes of VL treatment during pregnancy is rarely reported and under-researched. The reported articles were mainly case reports and case series and the reported information was often incomplete. From the studies identified, it is difficult to derive a generalisable information on outcomes for pregnant women and babies, although reported data favours the usage of liposomal amphotericin B for the treatment of VL in pregnant women.

## Author summary

Visceral Leishmaniasis (VL) is a neglected tropical disease with an estimated incidence of 50,000 to 90,000 cases in 2019. Women who are susceptible to becoming pregnant or those who are pregnant and lactating are regularly excluded from clinical studies of VL. A specific concern of public health relevance is the little knowledge of the consequences of VL and its treatment on the mother and the foetus. We did a systematic review of all published literature with an overarching aim of identifying cases of VL in pregnancy and assessing the risk-benefit balance of antileishmanial treatment to the pregnant women and the child. We identified a total of 72 records (1926–2020) describing 451 VL cases in pregnant women. In 398, infection was identified during pregnancy of whom 202 received Liposomal Amphotericin B (L-AmB) and 93 received pentavalent antimony (PA). In studies that reported maternal outcomes, reports of maternal death abortion/spontaneous abortion, and miscarriages were proportionally lower among those who received L-AmB compared to PA (no formal test of significance carried out). A total of 26 cases of confirmed, probable or suspected cases of vertical transmission were identified and the median time to detection was 6 months (range: 0–18 months). Our review brings together scattered observations of VL in pregnant women in the clinical literature and clearly highlights that the disease in pregnancy is under-reported and under-studied. The collated evidence derived mainly from case reports and case series indicate that L-AmB has a favourable safety profile than the antimony regimen and should be the preferred treatment for VL during pregnancy.

## Introduction

Visceral Leishmaniasis (VL) is a neglected tropical disease caused by the parasite *Leishmania donovani* in Asia and Africa and *Leishmania infantum* in the Mediterranean Basin, the Middle

East, Central Asia, South America, and Central America [1]. The disease is transmitted by female sandflies with an estimated 50,000 to 90,000 cases in 2019 [2]. India, Sudan, Brazil, Ethiopia, Kenya, and South Sudan accounted for more than 80% of the total VL cases reported to the WHO in 2019 [3].

The main clinical features of VL include persistent fever, splenomegaly, weight loss, and anaemia [1]. If untreated, the disease is generally fatal within 2 years [1]. Among those with clinical suspicion of the disease, diagnosis of VL is confirmed through demonstration of the parasites in a tissue aspirate obtained from the spleen, bone marrow, or lymph node. The accuracy of the diagnosis is sample dependent with a sensitivity greater than 90% for spleen specimen (gold standard approach) and 50–80% for bone marrow sample, but a splenic aspirate carries a risk of haemorrhage in one per 1,000 procedures [1]. Lipososmal amphotericin B (L-AmB) is the drug of choice for the treatment of VL in the Indian sub-continent, while a combination of pentavalent antimony and paromomycin is used in Eastern Africa. Blood transfusion may be required before initiation of treatment [4–6] or during the treatment or post-treatment follow-up period [7, 8]. In the clinical literature, the disease is predominantly described among males [9, 10].

A specific concern of public health relevance is the little knowledge of the clinical aspects of VL and treatment outcomes in pregnant and lactating women [11]. VL in pregnancy presents a number of challenges. During pregnancy, the use of splenic aspirate poses additional risk for the foetus. Pregnant women are also more likely to present with more severe anaemia than non-pregnant women and they are at an increased risk of requiring blood transfusion [11, 12]. During pregnancy, the optimal case management must take into account the consequences of the disease and the therapeutic intervention on the pregnant woman and the foetus [13]. Of note, except amphotericin B, all other available drugs are either contraindicated or subjected to restricted use in pregnant and lactating women and in women of child-bearing age (Table 1) [14–16]. Further complexities arise from potential vertical transmission of the disease either congenitally or through transplacental infection as a result of blood exchange during labour [17–19]. Such vertical transmission can induce *in utero* death or can be potentially deleterious to the foetus and infant [16, 20, 21]. While vertical transmission of VL is well-studied and established in animal studies, reports in humans are sporadic with observations of clinical manifestation several months post-partum [21–24].

The regulatory restrictions and limited evidence on safety of antileishmanial chemotherapeutics on the mother-foetus pair meant that historically clinicians had to rely on personal experience or limited published case-reports to make a decision regarding treatment. This led to some clinicians delaying the treatment of pregnant women until after delivery, especially when the case was detected closer to the due date [25, 26]. Others had treated them when the adjudicated risk of VL to the mother outweighed the risk posed by the drug to the mother-foetus pair [27]. Currently liposomal amphotericin B (L-AmB) remains the preferred regimen for the treatment in pregnancy (Table 1). However, pregnant and lactating women are regularly excluded from clinical studies [10] and are considered "therapeutic orphans" [28]. In studies that enrol females of childbearing age, counselling measures are usually set in place to inform the patients regarding the potential teratogenic harms of study drugs and either adoption of suitable contraception methods or observance of abstinence is mandatory [10]. In regular clinical practice and non-clinical trial settings, pregnancy tests and counselling however, might not be done routinely. A study conducted in South Asia found that only one in every six doctors ruled out pregnancy before prescribing miltefosine, a medication which has been contraindicated in pregnancy [29].

Finally, there is a lack of active pregnancy registries for most of the antileishmanial drugs except for miltefosine. In the context of Impavido (Profounda Inc.), the commercial name of

**Table 1. Antileishmanial usage during pregnancy.**

| Drug | Indication | FDA category (reviewed in [33, 34])[a] |
|---|---|---|
| Pentavalent antimony: Pentostam (Sodium Stibogluconate) | "Pentavalent antimonials are less safe in pregnancy, as they can result in spontaneous abortion, preterm deliveries and hepatic encephalopathy in the mother and vertical transmission" –Source: WHO-2010 [13] "Do not give during pregnancy unless there is no choice" –Source: South Sudan country guidelines [35] | C (Risks cannot be ruled out) |
| Amphotericin B deoxycholate | "Amphotericin B deoxycholate and lipid formulations are the best therapeutic options for visceral leishmaniasis. No abortions or vertical transmission have been reported in mothers treated with liposomal amphotericin B" –Source: WHO-2010 [13] | B (No evidence of risk from existing studies) |
| Liposomal amphotericin B (AmBisome) | "Animal studies do not indicate direct or indirect harmful effects with respect to reproductive toxicity. The safety of AmBisome in pregnant women has not been established. Systemic fungal infections have been successfully treated in pregnant women with conventional amphotericin B without obvious effect on the foetus, but the number of cases reported is insufficient to draw any conclusions on the safety of AmBisome in pregnancy. AmBisome should only be used during pregnancy if the possible benefits to be derived outweigh the potential risks to the mother and foetus. It is unknown whether AmBisome is excreted in human breast milk. A decision on whether to breastfeed while receiving AmBisome should take into account the potential risk to the child as well as the benefit of breast feeding for the child and the benefit of AmBisome therapy for the mother" –Source: The EMC [36] "Amphotericin B deoxycholate and lipid formulations are the best therapeutic options for visceral leishmaniasis. No abortions or vertical transmission have been reported in mothers treated with liposomal amphotericin B" –Source: The WHO-2010 [13] This is the first line therapy for treatment against pregnancy in Kenya [37], Ethiopia [38], Somalia [39], Republic of Sudan [40], South Sudan [35], Uganda [41], and Brazil [42, 43] | B (No evidence of risk from existing studies) |
| Pentamidine | Contraindicated during the first trimester of pregnancy –Source: WHO-2010 [13] | C (Risks cannot be ruled out) |
| Miltefosine (IMPAVIDO) | Contraindicated in pregnancy "IMPAVIDO may cause fetal harm. Fetal death and teratogenicity, occurred in animals administered miltefosine at doses lower than the recommended human dose. Do not administer IMPAVIDO to pregnant women. Obtain a serum or urine pregnancy test in females of reproductive potential prior to prescribing IMPAVIDO. Advise females of reproductive potential to use effective contraception during therapy and for 5 months after therapy" –Source: Prescribing information for IMPAVIDO [44] "Miltefosine is potentially embryotoxic and teratogenic and should not be used during pregnancy. Women of child-bearing age should be tested for pregnancy before treatment and use effective contraception for 3 months after treatment" –Source: WHO-2010 [13] | D (Positive evidence of risk) |
| Paromomycin (aminosidine) | "Ototoxicity in the fetus is the main concern. Insufficient data are available on the use of paromomycin in pregnant women" –Source: WHO-2010 [13] "Paromomycin crosses the placenta and can cause renal and auditory damage in the unborn child. Paromomycin is excreted in breast milk and adverse effects in the breastfed infant cannot be excluded." –Source: National guidelines of Kenya-2017 [37] | No category assigned |

[a] Silva et al (2013)[33] review is in context of treatment of pregnancy in American tegumentary leishmaniasis. The following definition of FDA category is presented: Category B—remote possibility of foetal harm; animal studies showed no risk to the foetus; there are no studies in humans; Category C- possible harm to the foetus; insufficient controlled studies in humans and animals [33].

miltefosine registered to the US medicines regulatory agency (US Food Drug Administration), a pregnancy registry was established to fulfil post marketing requirements [30, 31]. The recruitment of pregnant women as part of the observational study started in 2015 and the study is expected to be completed in 2026, and is estimated to recruit 0–1 patients per year over the 10 year study period, hence unlikely to generate a large volume of new safety data

[30]. There are no other active pregnancy registries on exposures to VL treatments from which to derive information on consequences on gestation, mother, foetus, and the newborn. Therefore, to understand the risks and benefits of treatment to the pregnant woman and the child, one must turn to the published literature.

The most comprehensive reviews on VL in pregnant women were conducted in the mid 2000s [21, 32]. We therefore conducted a systematic review of all published literature with an overarching aim of identifying cases of VL in pregnancy. The specific objectives were to assess the risk-benefit balance of antileishmanial therapies to the pregnant woman and the child and to identify the cases of vertical transmission. The review was not limited by language or any interventions.

## Material and methods

### Literature search

A review of all published literature was undertaken on 26th of March 2020 to identify records describing VL in pregnant women or any reports of vertical transmission of the disease in humans by searching the following clinical databases: Ovid MEDLINE; Ovid Embase; Cochrane Database of Systematic Reviews; Cochrane Central Register of Controlled Trials; World Health Organization Global Index Medicus: LILACS (Americas); IMSEAR (South-East Asia); IMEMR (Eastern Mediterranean); WPRIM (Western Pacific); ClinicalTrials.gov; and the WHO International Clinical Trials Registry Platform (ICTRP). The systematic review was conducted in accordance with the Preferred Reporting Items for Systematic-Reviews and Meta-Analyses (PRISMA) guidelines (S1 Text)[45]. In addition, full text screening of the publications indexed in the Infectious Diseases Data Observatory (IDDO) clinical trials library was carried out to identify any description of VL in pregnant women [46]. The references of all included publications were further checked to identify any relevant articles. This review is not registered and the protocol describing the search strategy including search strings, search dates and eligibility criteria for screening is presented in supplemental file (S2 Text).

### Study screening

Study screening was carried out in two stages to identify the studies fulfilling the inclusion and exclusion criteria (S2 Text): title and abstract screening (stage I) and then full-text screening (stage II). As reports on VL in pregnancy are sparse, articles meeting minimal inclusion criteria and non-primary research articles such as opinion pieces, clinical guidelines, textbooks, chapters, correspondences, reports of accidental inclusion in trials, or case reports of unplanned pregnancies during the study follow-up were also considered for comprehensiveness. No restrictions were applied regarding study design, follow-up duration, sample size, region, or the treatment regimen for eligibility of inclusion in this review. Articles that were not in English language (Spanish, Portuguese, Korean, and German) were reviewed using google translation (https://translate.google.co.uk/).

The articles were screened against eligibility criteria by a single reviewer (PD). A second reviewer was consulted (SSP) when the first reviewer could not reliably assess the eligibility. The first reviewer (PD) extracted data from all the eligible records and it was verified by the second reviewer (SSP) (who was not blinded) on all publications included in the review. Any discrepancy in the extracted information was flagged by the second reviewer and the differences were resolved through consensus. Screening and data extraction was carried out on a prospectively designed Excel database (S1 and S2 Data).

### Data extraction

The following bibliographic information were extracted: study title, name of the first author, year of publication, name of the study site and country. The following maternal and child characteristics were extracted: age of the pregnant woman, period of gestation (or trimester), treatment administered including drug dosage, follow-up duration, the outcome of the treatment for pregnant woman (cured, relapsed, death), and foetal outcomes (abortion, stillbirth, premature birth, healthy born, vertical transmission).

### Definitions

The records were classified as: case report/case series, prospective cohort or retrospective cohort studies. Records describing one or a small group of patients included as a part of prospective (or retrospective) studies in which VL in pregnancy was not of primary focus were considered as case report/case series. Similarly, studies that described a cohort of pregnant women without selection of a non-pregnant comparator group were also considered as case series. Countries were classified into sub-regions according to United Nations designation of geographical regions [47].

### Data analysis

Since the majority of the studies included were either case reports or case series, analysis of data was restricted to presentation of descriptive statistics and meta-analysis was not carried out. Descriptive summaries were presented for the characteristics of the studies included in the review, characteristics of pregnant woman (trimester, gestational age), treatment regimen including dosage and duration, clinical outcomes on the pregnant woman and the child. Graphics were generated using R software [48].

### Assessment of risk of bias

The risk of bias in case report/case series was assessed using a checklist proposed in Murad-2018 [49]. The following domains were assessed: patient selection, ascertainment of exposure, follow-up of pregnant women until delivery to record pregnancy related outcomes, and completeness of reporting of treatment and outcome status. Methodology and definitions adopted for assessment of risk of bias is presented in S1 Table. Briefly, bias in ascertainment of exposure was considered high if VL diagnosis was based solely on clinical features. A single case report was considered to be at a high risk of selection bias whereas a series of cases selected based on an audit of complete records over a study period was considered to be at a low risk of selection bias. If a study followed-up the pregnant women until birth and reported outcomes of the treatment for the mother–foetus pair, then follow-up was considered adequate to assess pregnancy-related outcomes. Bias in reporting of results was assessed based on the completeness of reporting on treatment status and clinical outcomes (See S1 Table for further explanations). For cohort studies (prospective or retrospective), risk of bias was assessed using The Newcastle-Ottawa scale. Two authors (PD, SSP) assessed the risk of bias in the studies included.

### Results

We identified 395 records from the literature searches up until 26[th] of March 2020, of which 272 were unique after removing duplicate entries. Of the 272 unique records, 99 were excluded at title and abstract screening stage leaving 173 records for full-text assessment of which 54 met the eligibility criteria for inclusion in the review (Fig 1). An additional 18 records were identified by searching the references of the eligible records or through personal

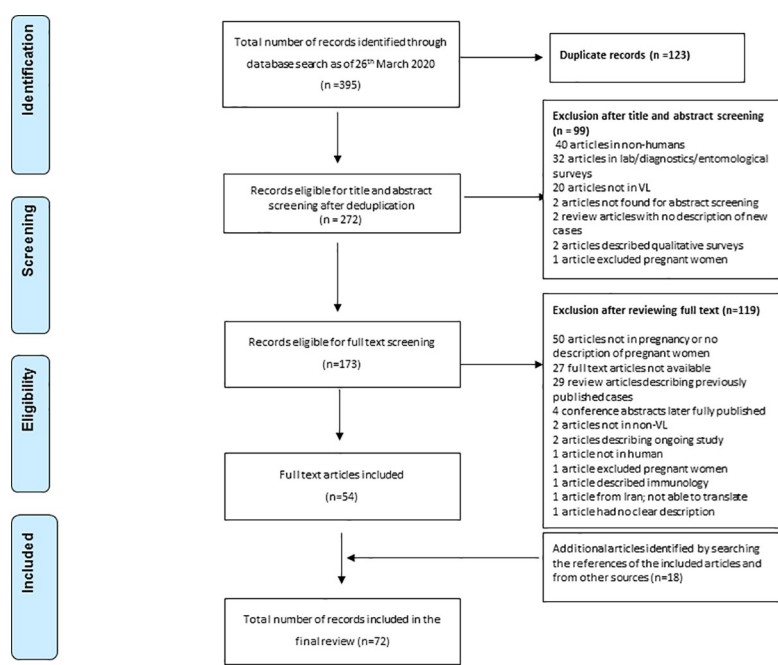

**Fig 1. Preferred Reporting Items for Systematic Reviews and Meta-Analyses (PRISMA) flow diagram of publications screened.**

communication. A total of 72 records published from 1926 through 2020 were included in this review of which 71 were case-reports or case-series and one was a retrospective cohort study with non-pregnant patient as a comparative group (Table 2). Further details on the studies included in this review are presented in supplemental files (S1 and S2 Data).

## Spatial and temporal distribution

A total of 21 (29.2%) records were from Europe (1926–2019), 23 (31.9%) from Southern Asia (1955–2018), 13 (18.1%) from South America (1986–2019), 8 (11.1%) from Northern Africa (1988–2018), 5 (6.9%) from Eastern Africa (1984–2020), and 1 (1.4%) record each were from Eastern Asia (1928) and Western Asia (1979). There were 18 (25.0%) records from India, 13 (18.1%) from Brazil, 8 (11.1%) from Sudan, and further breakdown by country is presented in Fig 2 (left panel). There were 64 (88.9%) records in English language, 7 (9.7%) in Portuguese, and 1 (1.4%) in French. The 72 records included in this review described 451 cases of VL in pregnant women, of whom 159 (35.3%) were from Sudan, 113 (25.1%) from South Sudan, 84 (18.6%) from India, 23 (5.1%) from Bangladesh, 20 (4.4%) from Brazil, 12 (2.7%) from Italy, 10 (2.2%) from Uganda, and the rest of the breakdown is presented in Fig 2 (right panel). Overall, there were 10 (13.9%) records published from 1926 through 1990, 14 (19.4%) from 1991 through 2000, 22 (30.6%) from 2001 through 2010, and 26 (36.1%) from 2011 through 2020 (Table 2).

## Treatment regimens

Of the 451 pregnant women identified, the disease was detected during pregnancy in 398 (88.2%), retrospectively confirmed after giving birth in 52 (11.5%), and the time of identification was not clear in one (0.2%). Of the 398 pregnant women whose infection was identified during pregnancy, 346 (86.9%) received a treatment, 3 (0.8%) were untreated, and the

**Table 2. Description of reported cases of visceral leishmaniasis in pregnant or lactating women.**

| Author-year | Country | Time of detection/ description | Number of pregnant woman/ women | Trimester | Description of treatment in pregnancy | Pregnancy outcome |
|---|---|---|---|---|---|---|
| Low-1926 [19] | UK | During pregnancy | 1 | 3rd | Urea stibamine | Normal delivery |
| Hindle-1928 [66] | China | Retrospectively suspected | 1 | Not applicable (retrospective) | No information | No description |
| Hindle-1928 [66] | China | Retrospectively suspected | 1 | Not applicable (retrospective) | No information | No description |
| Banerji-1955 [56] | India | During pregnancy | 1 | 2nd | Treated with 10 intravenous Urea stibamine | Remission of fever |
| El-Saaran-1979 [54] | UAE | During pregnancy | 1 | 2nd | Pentostam: 6 ml intravenous dose daily for ten days, at intervals of ten days to a total of 180 ml | Did not recover; splenectomy performed after birth |
| Rees-1984 [67] | Kenya | During pregnancy | 1 | No information | Pentostam | No information |
| Blanc-1984 [68] | France | Retrospectively identified | 1 | Not applicable (retrospective) | Treatment with N-methylglucamine (antimony) for 10 days | Normal delivery |
| Badaro-1986 [69] | Brazil | During pregnancy | 1 | 3rd | Untreated (posthumous diagnosis) | Mother died 5 weeks later |
| Mittal-1987 [62] | Indian | During pregnancy | 1 | 3rd | Drug name not stated | Normal term delivery |
| Nyakundi-1988 [70] | Sudan | During pregnancy | 1 | Not clear | Unclear | Premature birth at 6 months of gestation |
| Yadav-1989 [71] | India | Retrospectively identified (symptoms shown during pregnancy) | 1 | 2nd | Not treated (herbal medicine was given) | Normal and uneventful delivery |
| Aggarwal-1991 [72] | India | During pregnancy | 1 | 2nd (6 months) | Sodium antimony gluconate | Healthy baby delivered at full term |
| Aggarwal -1991 [72] | India | During pregnancy | 2 | No information | No information | No information |
| Elamin1992 [17] | Sudan | During pregnancy | 1 | 3rd | Drug name not stated | Normal delivery |
| Eltoum-1992 [18] | Sudan | During pregnancy | 1 | 2nd | SSG: 10 mg/kg/daily for 30 days | Normal delivery |
| Eltoum-1992 [18] | Sudan | During pregnancy | 1 | 2nd | Not clear | Abortion of a female foetus (5 months old) |
| Seaman-1993 [27] | Sudan | During pregnancy | 3 | Not clear | SSG 20 mg/kg/day for 30 days | No information |
| Seaman-1993 [27] | Sudan | During pregnancy | 3 | Not clear | SSG + aminosidine (20 mg/kg/ day SSG for 17 days + 15 mg/kg of aminosidine for 17 days) | No information |
| Thakur-1993 [73] | India | During pregnancy | 1 | 2nd | Amphotericin B (1 mg/kg body weight daily starting with 0.5 mg/ kg body weight till a total dose of 20 mg/kg) | Normal delivery |
| Thakur-1993 [73] | India | During pregnancy | 1 | 2nd | Amphotericin B (1 mg/kg body weight daily starting with 0.5 mg/ kg body weight till a total dose of 20 mg/kg) | Normal delivery |
| Thakur-1993 [73] | India | During pregnancy | 1 | 1st | Amphotericin B (1 mg/kg body weight daily starting with 0.5 mg/ kg body weight till a total dose of 20 mg/kg) | Normal delivery |
| Thakur-1993 [73] | India | During pregnancy | 1 | 1st (febrile episode 40 days after conception) | Amphotericin B (1 mg/kg body weight daily starting with 0.5 mg/ kg body weight till a total dose of 20 mg/kg) | Normal delivery |
| Thakur-1993 [73] | India | During pregnancy | 1 | 2nd | Amphotericin B (1 mg/kg body weight daily starting with 0.5 mg/ kg body weight till a total dose of 20 mg/kg) | Normal delivery |

(*Continued*)

**Table 2.** (Continued)

| Author-year | Country | Time of detection/ description | Number of pregnant woman/ women | Trimester | Description of treatment in pregnancy | Pregnancy outcome |
|---|---|---|---|---|---|---|
| Giri-1993 [74] | India | During pregnancy | 1 | 3rd | Amphotericin B | Normal term delivery |
| Gradoni-1994 [75] | Italy | During pregnancy | 1 | 2nd | L-AmB total dose of 18 mg/kg | Normal delivery |
| Gradoni-1994 [75] | Italy | Retrospectively identified (treated after delivery) | 1 | Not applicable (retrospective) | Untreated (diagnosed after birth); treated with 18 mg/kg/day PA after birth | Normal delivery |
| Jeronimo-1994 [76] | Brazil | During pregnancy | 1 | No information | Meglumine antimoniate (20 mg/kg/day for 20 days) | No information |
| Bano -1994 [61] | India | During pregnancy | 1 | 1st (12 weeks pregnant) | Trimothoprim, sulphadiazine and tinidazole | Normal full term delivery of a healthy baby |
| Utili-1995 [23] | Italy | During pregnancy | 1 | 2nd | Meglumine antimoniate (12 mg/kg for 20 days) | Normal term birth (patient delivered a baby weighing 4.2 kg at 41 weeks of pregnancy) |
| Sharma-1996 [77] | India | Retrospectively identified | 1 | Not applicable (retrospective) | Untreated | Normal delivery |
| Thakur-1998 [78] | India | During pregnancy | 1 | No information | Amphotericin B deoxycholate (total dose 20 mg/kg) | Normal delivery |
| Thakur-1998 [78] | India | During pregnancy | 1 | No information | Amphotericin B deoxycholate (total dose 20 mg/kg) | Normal delivery |
| Thakur-1999 [5] | India | During pregnancy | 1 | No information | Amphotericin B (total dose 20 mg/kg) | Normal delivery |
| Thakur-1999 [5] | India | During pregnancy | 1 | No information | Amphotericin B (total dose 20 mg/kg) | Normal delivery |
| Thakur-1999 [5] | India | During pregnancy | 1 | No information | Amphotericin B (total dose 20 mg/kg) | Normal delivery |
| Thakur-1999 [5] | India | During pregnancy | 1 | No information | Amphotericin B (total dose 20 mg/kg) | Normal delivery |
| Meinecke-1999 [79] | Germany | Retrospectively identified | 1 | Not applicable (retrospective) | Untreated (retrospective identification) | Complicated pregnancy with febrile gastroenteritis; birth weight was 3,720 g |
| Vianna-2001 [80] | Brazil | Retrospectively identified | 2 | Not applicable (retrospective) | Untreated (retrospective identification) | One preterm birth |
| Kumar-2001 [25] | India | During pregnancy | 1 | 3rd | Untreated (treatment deferred until birth) | Intrauterine growth retardation; small for age baby; Emergency C-section required |
| Dereure-2003 [81] | France | During pregnancy (routine check-up) | 1 | 2nd | L-AmB (3 mg/kg daily for five days; a 6th injection 10 days later) | Normal term birth |
| Caldas-2003 [82] | Brazil | During pregnancy | 1 | 1st | Amphotericin B (1mg/kg for 14 days) | Normal term birth |
| Silveira-2003 [55] | Brazil | During pregnancy | 1 | 2nd | Meglumine antimoniate (850mg/day for 20 days) | Premature birth |
| Pagliano-2003 [83] | Italy | During pregnancy | 2 | No information | L-AmB | Normal delivery |
| Kumar-2004 [84] | Iran | Retrospectively identified (after death of mother-child) | 1 | 3rd | Untreated (posthumous diagnosis) | Death |
| Pagliano-2005 [21] | Italy | During pregnancy | 1 | Not clear | L-AmB (3 mg/kg at days 1–5 & 3 mg/kg at day 10) | Healthy term birth |
| Pagliano-2005 [21] | Italy | During pregnancy | 1 | Not clear | L-AmB (3 mg/kg at days 1–5 & 3 mg/kg at day 10) | Healthy term birth |
| Pagliano-2005 [21] | Italy | During pregnancy | 1 | Not clear | L-AmB (3 mg/kg at days 1–5 & 3 mg/kg at day 10) | Healthy term birth |
| Pagliano-2005 [21] | Italy | During pregnancy | 1 | Not clear | L-AmB (3 mg/kg at days 1–5 & 3 mg/kg at day 10) | Healthy term birth |

(*Continued*)

**Table 2.** (Continued)

| Author-year | Country | Time of detection/ description | Number of pregnant woman/ women | Trimester | Description of treatment in pregnancy | Pregnancy outcome |
|---|---|---|---|---|---|---|
| Pagliano-2005 [21] | Italy | During pregnancy | 1 | Not clear | L-AmB (3 mg/kg at days 1–5 & 3 mg/kg at day 10) | Healthy term birth |
| Figueiró Filho-2005 [85] | Brazil | During pregnancy | 1 | 3rd | L-AmB (1 mg/kg/day for 21 days) | Normal birth at 38 weeks with baby weighing 2,995g |
| Mueller-2006 [59] | Sudan | During pregnancy | 23 | 11 in 1st; 8 in 2nd; 4 in 3rd | SSG: 20 mg/kg for 30 days | 13 spontaneous abortion during days 13 to 30 of SSG; 1 spontaneous abortion prior to treatment; 1 healthy baby born; remaining 8 still pregnant at discharge |
| Mueller-2006 [59] | Sudan | During pregnancy | 4 | 2 in 2nd; and 2 in 3rd | L-AmB + SSG (AmBisome 3–7 mg/kg daily on days 1, 6, 11 and 16 (or on days 1,2,3,4,10, and 15), followed by 20 mg/kg SSG IM once daily for 30d) | 1 healthy baby born; remaining 3 were still pregnant at discharge |
| Mueller-2006 [59] | Sudan | During pregnancy | 12 | 2 in1st; 6 in2nd; 4 in 3rd | L-AmB (AmBisome 3–7 mg/kg daily on days 1, 6, 11 and 16 (or on days 1,2,3,4,10, and 15) | Premature birth (n = 1); two healthy babies; remaining 9 still pregnant at discharge |
| Boehme-2006 [86] | Germany | Possibly before pregnancy | 1 | Not applicable (retrospective) | No treatment given (retrospective speculation) | Spontaneous birth at 39 weeks of gestation and healthy baby delivered |
| Vieira-2007 [64] | Brazil | During pregnancy | 1 | 3rd | Untreated | Baby died 2 months after birth |
| Topno-2008 [88] | India | During pregnancy | 1 | 2nd | Amphotericin B (15 infusions of 1 mg/kg) | Normal term birth |
| Topno-2008 [88] | India | During pregnancy | 1 | 2nd | Amphotericin B (15 infusions of 1 mg/kg) | Normal term birth |
| Topno-2008 [88] | India | During pregnancy | 1 | 3rd | Amphotericin B (15 infusions of 1 mg/kg) | Normal term birth |
| Topno-2008 [88] | India | During pregnancy | 1 | 3rd | Amphotericin B (15 infusions of 1 mg/kg) | Normal term birth |
| Figueiró-Filho-2008 [57] | Brazil | During pregnancy | 1 | Not clear (between 28 ± 7.8 weeks) | Amphotericin B deoxycholate (1 mg/kg/day for 20 days) | One maternal death after 7 days of treatment due to haemorrhagic complications occurring after delivery |
| Figueiró-Filho-2008 [57] | Brazil | During pregnancy | 1 | Not clear (between 28 ± 7.8 weeks) | Amphotericin B deoxycholate (1 mg/kg/day for 20 days) | No information |
| Figueiró-Filho-2008 [57] | Brazil | During pregnancy | 1 | Not clear (between 28 ± 7.8 weeks) | L-AmB (3 mg/kg/day for 20 days) | No information |
| Figueiró-Filho-2008 [57] | Brazil | During pregnancy | 1 | Not clear (between 28 ± 7.8 weeks) | L-AmB (3 mg/kg/day for 20 days) | No information |
| Figueiró-Filho-2008 [57] | Brazil | Retrospectively confirmed (After giving birth) | 1 | Not clear (between 28 ± 7.8 weeks) | Untreated during pregnancy (Diagnosed after birth and given SSG: 20 mg/kg/day for 20 days) | No information |
| Lorenzi-2008 [89] | UK | Retrospectively identified | 1 | Not applicable (retrospective) | Untreated during pregnancy (Diagnosed after 3 years and treated with amphotericin B) | Miscarriage |
| Adam-2009 [12] | Sudan | During pregnancy | 42 | 9 in 1st; 21 in 2nd; 12 in 3rd | SSG: 20 mg/kg SSG once daily IM for 30 days | Miscarriage in first trimester (n = 2); death due to hepatic encephalopathy (n = 4); preterm birth (n = 2) |
| Mueller-2009 [63] | Uganda | During pregnancy | 10 | No information | PA or amphotericin b deoxycholate | 2 maternal deaths |

(*Continued*)

**Table 2.** (Continued)

| Author-year | Country | Time of detection/ description | Number of pregnant woman/ women | Trimester | Description of treatment in pregnancy | Pregnancy outcome |
|---|---|---|---|---|---|---|
| Papageorgiou-2010 [51] | Greece | Confirmed few days before labour | 1 | 3rd | L-AmB (4 mg/kg on 6 consecutive days and repeated doses at days 14 and 21) | No information |
| Miah-2010 [26] | Bangladesh | During pregnancy | 11 | 2nd | SAG (20 mg/kg for 30 days) | Abortion (n = 11) |
| Miah-2010 [26] | Bangladesh | During pregnancy | 5 | 3rd | SAG (20 mg/kg for 30 days) | Good outcome (n = 5) |
| Zinchuk -2010 [52] | Ukraine | During pregnancy | 1 | 3rd | L-AmB (3 mg/kg days 1–5 followed by a single dose 3 mg/kg on day 10) | Delivery by elective C-section at 38 weeks of gestation; baby birth weight of 2,800g |
| Sinha-2010 [90] | India | During pregnancy | 3 | No information | L-AmB (5 mg/kg on days 0, 1, 4, and 9) | Not described (successful treatment) |
| Haque-2010 [91] | Bangladesh | Retrospectively identified | 1 | Not applicable (retrospective) | Untreated during pregnancy | Vertical transmission identified at 15 days of birth |
| Ritmeijer-2011 [92] | Ethiopia | During pregnancy | 1 | No information | L-AmB (6 infusions of 5 mg/kg) | Good response to treatment |
| Ritmeijer-2011 [92] | Ethiopia | During pregnancy | 1 | No information | L-AmB (6 infusions of 5 mg/kg) | Good response to treatment |
| Sinha-2011 [60] | India | During pregnancy | 3 | No information | Paromomycin (11 mg/kg/day for 21 days) | Normal delivery |
| Pilaca-2011 [93] | Albania | Retrospectively identified (diagnosis was made on the day the baby was born) | 1 | 3rd | Untreated [After giving birth: Glucantime for 28 days. The baby was not fed by his mother's breast. PA given as L-AmB was not available] | Preterm birth |
| Damodaran-2012 [94] | UK | Retrospectively identified | 1 | Not applicable (retrospective) | No information | Vertical transmission (suspected) at 15 months |
| Lima-2013 [95] | Brazil | Retrospectively identified (After 5 days of giving birth) | 1 | Not applicable (retrospective) | Untreated during pregnancy (diagnosed after birth); After diagnosis L-AmB (total dose of 20 mg/kg over 5 days) | Preterm vaginal birth; mother died on the 32nd day after birth |
| Lima-2013 [95] | Brazil | During pregnancy | 1 | 1st | L-AmB | No information |
| Mescouto-Borges-2013 [96] | Brazil | Retrospectively identified (After giving birth) | 1 | 2nd | Untreated (diagnosed after birth); Amphotericin b deoxycholate 1 mg/kg followed by IV L-AmB 3mg/kg/day | Acute foetal distress requiring C-section delivery; extremely premature birth (1,170g) |
| Mescouto-Borges-2013 [96] | Brazil | During pregnancy | 1 | 2nd | Untreated (diagnosed after birth); IV L-AmB given at 3 mg/kg/day for 7d | Acute foetal distress requiring C-section delivery; premature birth |
| Milosevic-2013 [97] | Serbia | Retrospectively identified (After 31 days of giving birth) | 1 | Not applicable (retrospective) | Untreated (diagnosed after birth) | Normal vaginal delivery |
| Salih-2014 [98] | Sudan | During pregnancy | 23 | No information | L-AmB (30 mg/kg divided into 10 IV infusions of 3 mg/kg) | No information |
| Burza-2014 [99] | India | During pregnancy | 49 | No information | AmBisome | No information |
| Bode-2014 [100] | Germany | Not clear | 1 | No information | No information | Vertical transmission at 8 months |
| Llamazares-2014 [101] | Spain | Retrospectively identified | 1 | Not applicable (retrospective) | Untreated during pregnancy | Normal delivery |
| Rahman-2014 [102] | Bangladesh | During pregnancy | 1 | No information | L-AmB | Stillbirth baby |
| Colomba-2015 [103] | Italy | Retrospectively identified (After 4 days of giving birth) | 1 | After delivery | Untreated: (treated with L-AmB 3 mg/kg/day on days 1–5 and on day 10) | No information |
| Pawar-2015 [104] | India | During pregnancy | 1 | 2nd | Amphotericin B deoxycholate (later switched to liposomal preparation to minimise nephrotoxicity) | Full term normal vaginal delivery at 38 weeks of gestation |

*(Continued)*

**Table 2.** (Continued)

| Author-year | Country | Time of detection/ description | Number of pregnant woman/ women | Trimester | Description of treatment in pregnancy | Pregnancy outcome |
|---|---|---|---|---|---|---|
| Kumar-2015 [105] | India | Retrospectively identified (After 5 months of delivery) | 1 | 3rd | Untreated | Normal vaginal birth |
| AlmeidaSilva-2015 [50] | Brazil | During pregnancy | 1 | 1st | L-AmB | Foetal death |
| Basher -2017 [65] | Bangladesh | During pregnancy | 5 | No information | One untreated; One was treated with L-AmB | Untreated mother died |
| Kimutai-2017 [58] (personal communication with Dr Alves) | East Africa | During pregnancy | 8 | No information | SSG+PM | Spontaneous abortion (n = 1) |
| Panagopoulos-2017 [106] | Greece | During pregnancy | 1 | 3rd | L-AmB (3 mg/kg/day for 5 days and on days 14 and 21) | Normal term birth |
| Adam-2018 [107] | Sudan | During pregnancy | 45 | Mostly 3rd | No information | 8 maternal death (6 in prenatal and 2 in postnatal); 37 survived; 30 were full term; 6 pre-term birth; 2 spontaneous abortion; 1 stillbirth |
| Goyal-2018 [108] (personal communication With Dr Alves) | India | During pregnancy | 2 | No information | Single dose AmBisome (10 mg/ kg) | No complications |
| Russo-2018 [109] | Italy | Retrospectively identified | 1 | Not applicable (retrospective) | No information | Vertical transmission |
| Cunha-2019 [110] | Brazil | During pregnancy | 1 | 3rd | L-AmB (3 mg/kg for 7 days) | Normal term birth without complications |
| Argy-2019 [53] | Brazil | During pregnancy | 1 | 3rd | L-AmB | Vertical transmission at birth |
| Parise-2019 [111] | France | Retrospectively identified | 1 | Not applicable (retrospective) | Untreated during pregnancy | Maternal death |
| Pekelharing-2020 [11] | South Sudan | During pregnancy | 87 | 26.8% in 1st 35.1% in 2nd 38.1% in 3rd | L-AmB (30mg/kg in 6 doses) | 1 maternal death; 16 adverse pregnancy outcomes |
| Pekelharing-2020 [11] | South Sudan | Retrospectively identified (two weeks post-partum) | 26 | Not applicable (retrospective) | L-AmB (30mg/kg in 6 doses) | 1 maternal death; 13 adverse pregnancy outcomes |

L- AmB = Liposomal amphotericin B; PA = pentavalent antimony; SSG = sodium stibogluconate; SAG = Sodium antimony gluconate; IV = intravenous; IM = intramuscular; PM = Paromomycin; Mueller-2007 [87] was identified after completion of the review and not included in this table - the study described 5 pregnant women treated with AmBisome: six doses of 2.5—8.2 mg/kg on days 1, 2, 3, 5, 10, 15 with no pregnancy outcomes not reported

treatment status was not clear in the remaining 49 (12.3%) (Table 2). The outcomes among 3 untreated cases and 346 who received a treatment during pregnancy is presented next.

**Liposomal amphotericin B (n = 202; 1994–2020).** Of the 202 pregnant women treated with Liposomal Amphotericin B, 124 (61.4%) were from Africa (2006–2020), 59 (29.2%) were from the Indian sub-continent (2010–2018), 13 (6.4%) were from Europe (2010–2018), and 6 (3.0%) were from South America (2005–2019).

Treatment was administered in 5 (2.5%) cases in the first trimester, 8 (4.0%) in the second trimester, 9 (4.5%) in the third trimester, while the time in pregnancy was not clear in 180 (89.1%). Survival status was not reported or was unclear in 26 (12.9%) pregnant women and

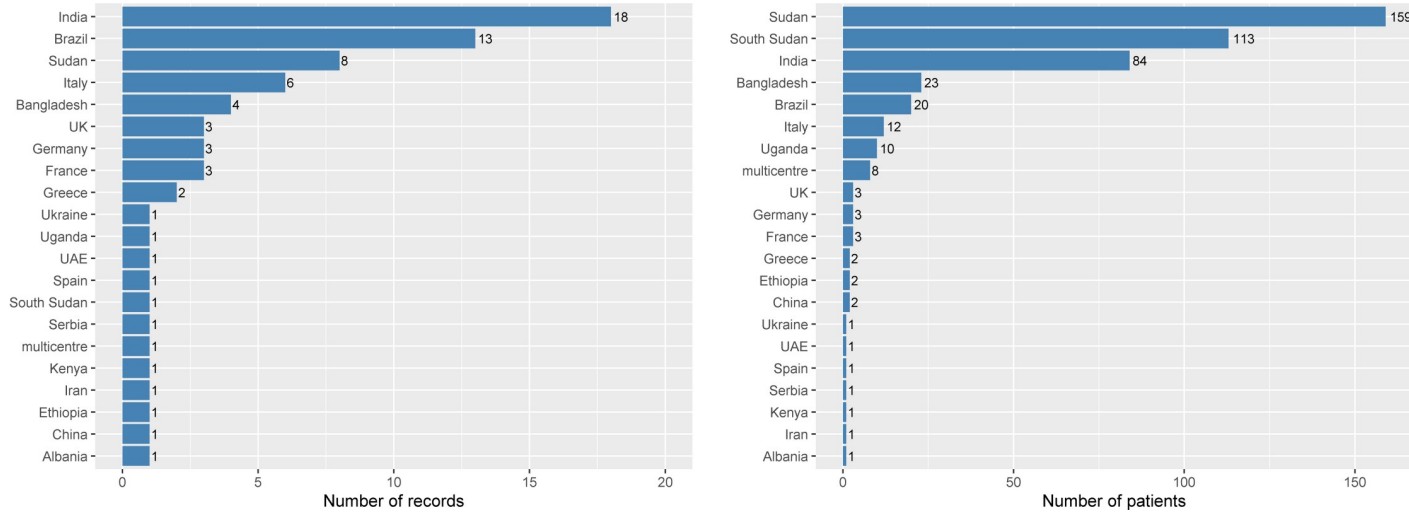

**Fig 2. Number of records and patients by country of origin.**

from the remaining 176, a total of four (2.3%, 4/176) maternal deaths were reported. There were five (2.8%) miscarriages, 1 (0.6%) stillbirth, and 1 (0.6%) premature birth (Table 3). One foetal death was reported from a pregnant woman in her first trimester from South America [50].

Three cases of vertical transmission were identified [51–53] (Tables 2 and 4). The first one was detected at 11 months after birth. The pregnant woman was admitted to a hospital few days before labour and was administered L-AmB at a daily dose of 4 mg/kg on 6 consecutive days and repeated doses at days 14 and 21 [51]. The treatment status on the baby was not available. For the

**Table 3. Reported unfavourable maternal outcomes after liposomal amphotericin B and antimony treatment among those who were treated during pregnancy.**

|  | Africa | Europe | South America | Southern Asia | Total |
|---|---|---|---|---|---|
| **Liposomal amphotericin B** |  |  |  |  |  |
| Total treated | 124 | 13 | 6 | 59 | 202 |
| Outcomes (reported in 176; unclear in 26) |  |  |  |  |  |
| Abortion/spontaneous abortion | - | - | - | - | - |
| Miscarriage | 5 | - | - | - | 5 |
| Premature birth | 1 | - | - | - | 1 |
| Stillbirth | - | - | - | 1 | 1 |
| Foetal death | - | - | 1 | - | 1 |
| Maternal death | 3 | - | - | 1 | 4 |
| **Pentavalent antimony** |  |  |  |  |  |
| Total treated | 70 | 2 | 2 | 19 | 93 |
| Outcomes (reported in 88; unclear in 5) |  |  |  |  |  |
| Abortion/spontaneous abortion | 13 | - | - | 11 | 24 |
| Miscarriage | 2 | - | - | - | 2 |
| Premature birth | 2 | - | 1[a] | - | 3 |
| Stillbirth | - | - | - | - | - |
| Foetal death | 2 | - | 1[a] | - | 3 |
| Maternal death | 4 |  |  |  | 4 |
| Other (required splenectomy after delivery) | - | - | - | 1 | 1 |

[a] Indicates the same patient (the baby was born prematurely and died a day after birth)

**Table 4. Details of confirmed, probable, or suspected cases of vertically transmitted visceral leishmaniasis.**

| Study[a] | Location | Case description |
|---|---|---|
| Low -1926 [19] | UK | A retrospective description of a child born to a mother who contracted the disease during pregnancy while residing in India and had given birth in the UK. |
| Hindle-1928 [66] | China | A four months old baby whose spleen puncture confirmed the presence of Leishmania parasites.<br>"The main interest of this case lies in the fact that it could not possibly have been exposed to the bites of sandflies, as their season ended approximately two months before the child was born. Although the mother showed no obvious signs of disease it is difficult of explanation except on the hypothesis of congenital transmission. Low and Cooke (1926) recorded a case of Indian Kala-azar in a child born in England, and there can be no doubt that in this patient the infection was derived from the mother who was also infected" |
| Hindle-1928 [66] | China | "Dr Marshall Hertig kindly informed me of a similar case at Hsii-Chowfu in which the patient, a five months old child, was successfully treated for Kala-azar at the local mission hospital. This infant also, from the date of its birth, could never have been exposed to the bites of sandflies" |
| Banerji-1955 [56] | India | Mother contracted kala-azar in the fifth month of pregnancy and suspected vertical transmission occurred when the child was 6 months old. |
| Blanc-1984 [68] | France | Mother with a subclinical infection during pregnancy with the disease detected within a month after delivery.<br>The child had a confirmed VL and was the first case reported in the hospital. The child never left the hospital and never came in contact of dogs thus suggesting that vertical transmission was the likely mode of transmission. |
| Mittal-1987 [62] | India | "An 11-months-old male infant admitted with symptoms that were later confirmed as VL. The baby's mother had also suffered from kala-azar while carrying this child. As the baby and his mother did not leave New Delhi, India, where the case was related, either during or after the delivery and the vector found in New Delhi was not competent to transmit leishmaniasis, the infant could not have been infected by the bite of a sandfly. It therefore seems most likely that he was congenitally exposed to kala-azar." |
| Nyakundi-1988 [70][b] | Kenya | "We recently treated a 4 months old male infant born prematurely on 18 June 1986, after 6 months gestation to a then febrile (para 6+3) mother diagnosed as having had kala-azar during pregnancy. Mother and infant were admitted to the Clinical Research Centre. Kenya Medical Research Institute, on 20 October-1986; when kala-azar was confirmed in the mother. This infant with congenital kala-azar was only the fourth and youngest patient with this disease ever reported in the world medical literature. The mode of infection in the baby could be (a) direct transmission from mother to offspring, (b) acquired in hospital, (c) acquired at the time of birth from perineal haemorrhages with swallowing of maternal blood or secretions or through the cord or skin abrasions, or (d) acquired congenitally from the mother through the placenta. Only the last of these possible modes of transmission is likely in view of the poor health of the infant from the 6th day of life, the mother's bad obstetric history, the hospital's high altitude which makes it unsuitable for sandfly transmission, and because the period that elapsed from birth to the appearance of symptoms was compatible with a congenital infection." |
| Yadav-1989 [71] | India | An 11-months male infant was admitted with kala-azar. The mother suffered from the disease during pregnancy. The mother from Bihar migrated to Delhi during first trimester. She showed signs of disease during sixth month of pregnancy. The most likely mode of infection was *in utero* transmission of the disease. |
| Eltoum-1992 [18] | Sudan | During an epidemic of visceral leishmaniasis in the Sudan, two cases of congenital kala-azar were seen.<br>The first child, whose mother had contracted kala-azar in southern Sudan, was born in Khartoum, where no transmission of leishmaniasis is currently occurring. At seven months, the child had fever, lymphadenopathy, and hepatosplenomegaly; leishmania parasites were detected in the bone marrow. The child died and an autopsy showed leishmania parasites in all tissues including the lungs, kidneys, and thymus.<br>In the second case, parasites were found in the placenta of a five-months-old foetus. |
| Elamin -1992 [17] | Sudan | A case of visceral leishmaniasis was described in a 6-weeks-old infant from southern Sudan who most likely got the infection through transplacental transmission. This was described as the first reported case of congenital kala-azar in Africa and the seventh in the global medical literature |
| Sharma-1996 [77] | India | "Thus, in all possibility, it was a case of congenital kala-azar acquired transplacental by the baby from a mother having subclinical kala-azar."The infection was possibly active when the child was 4 months of age and it was detected when the child was 18 months. |
| Meinecke-1999 [79] | Germany | A case of VL was reported in a child who had never left Germany. A nonvector transmission was suspected and therefore household contacts were examined. His mother was the only one who had a positive antibody titre against Leishmania donovani complex. She had travelled several times to endemic Mediterranean areas (Portugal, Malta, and Corse) before giving birth to the boy. But she had never been symptomatic for visceral leishmaniasis. Her bone marrow, spleen, and liver biopsy results were within normal limits. |
| Boehme-2006 [86] | Germany | A case of VL was reported in a German infant, who never had been to a VL endemic area. Most likely, the parasite was congenitally transmitted from the asymptomatic mother to her child. |
| Papageorgiou-2010 [51] | Greece | This article reports the first case of congenital VL in Greece. The mother of the infant was hospitalised a few days before labour because of anaemia and hepatosplenomegaly, and titres for Leishmania antibodies were positive. A bone marrow aspirate showed no evidence of malignancy, except from a slight decrease of myelopoiesis, erythropoiesis and thrombopoiesis. However, the promastigote form of Leishmania was found, and therefore, diagnosis of leishmaniasis was confirmed. |
| Haque-2010 [91] | Bangladesh | The article is the first report of vertical transmission of VL in Bangladesh. |

*(Continued)*

**Table 4.** (Continued)

| Study[a] | Location | Case description |
|---|---|---|
| Zinchuk -2010 [52] | Ukraine | An 8-months-old boy was diagnosed with visceral leishmaniasis in Ukraine, a non-endemic area. His mother had been treated for visceral leishmaniasis at 28–32 weeks gestation whilst working in Alicante, Spain and delivered her infant at 38 weeks gestation by elective caesarean section in Ukraine. The authors presumed that the infant's infection was as a result of vertical transmission. |
| Pilaca-2011 [93] | Albania | Leishmania amastigotes were detected in bone marrow biopsy of the mother. Two days later, premature birth was stimulated. After 2–3 months of the birth the baby was not well. After admitted to hospital, baby resulted positive for VL. He was treated with Glucantime and was cured after a scheme of two 14-day cycles with good outcome. |
| Damodaran-2012 [94] | UK | "A 15 months-old-girl, family from East Timor, referred from primary care with weight-loss and a non-healing skin ulcer. She appeared undernourished with pallor, pyrexia and hepatosplenomegaly. FBC showed pancytopenia. Bone marrow examination confirmed Leishmaniasis. Her mother had intrapartum Leishmaniasis. The child was born in United Kingdom with no history of foreign travel and responded well to treatment with AmBisome" |
| Mescouto-Borges-2013 [96] | Brazil | Two cases of congenitally transmitted visceral Leishmaniasis was reported. Both mothers had developed symptoms of VL during pregnancy. The diagnosis was made by visual examination of Leishmania parasites in bone marrow aspirates of the mothers and by detecting parasite DNA in bone marrow samples of the new-born children using polymerase chain reaction. |
| Bode-2014 [100] | Germany | "One infant girl (P8) had only been in an endemic area (Spain) *in utero*. Vertical transmission resulting in congenital visceral leishmaniasis must be assumed, as the mother, who remained clinically asymptomatic, was serologically positive. Diagnosis of visceral leishmaniasis was delayed for more than 3 weeks".<br>The girl had never been abroad after birth and the mother had positive Leishmania serology after a trip to Spain during pregnancy. |
| Kumar-2015 [105] | India | The authors presumed that the infant's infection was a result of vertical transmission. The authors presumed that the mother might have had subclinical infection and transmitted the disease to the offspring. |
| Basher -2017 [65] | Bangladesh | The authors described a mother who had died before VL treatment began. The mother died after giving birth to a foetus. Foetal part placenta was collected and upon examination using PCR, the sample tested positive for LD body. The authors reported that Kala-azar in the mother may have been the cause of the foetal wastage. |
| Russo-2018 [109] | Italy | The authors reported a vertically transmitted Leishmaniasis in a 6-months old girl whose parents were from Southern Italy. |
| Argy-2019 [53] | France | A case of transplacentally transmitted VL from the HIV-positive pregnant mother to the child. Few intracellular *Leishmania* amastigotes were found during the microscopic examination of the placenta confirmed by positive PCR results. |

VL = Visceral leishmaniasis; PCR = Polymerase chain reaction

[a] An article from Sudan (Adam 2009 [12]) described a case of a 2 months baby with parasites detected in lymph node. The article did not mention whether this could be a case of vertical transmission.

[b] In Nyakundi-1988[70], three cases of vertically transmitted VL in clinical literature were identified: Low and Cooke-1926[19]; Banerji-1955[56] and Napier-1946[112]. The first two reports are included in this table whereas we have decided not to include the last report as a case of vertical transmission as the original article could not be retrieved and case details couldn't be verified. The following description appears in Napier-1946 [112]: "Even in India kala-azar occurs among infants; we reported a case of an infant of less than eight months with well-developed kala-azar of about four months' duration". While it is clear that VL was identified when the infant was four months old, there is no further description of the case [112]. The brief description in Napier-1946[112] matches an earlier publication (Napier and Das Gupta-1928 [113]) in which the plausibility of vertical transmission was ruled out: "As the mother showed no sign of the disease at all it is extremely unlikely that the child was suffering from the disease at birth."

second case, vertical transmission was suspected at 8 months after birth. During weeks 31–32 of pregnancy, the pregnant woman was treated with a full course of L-AmB (3 mg/kg on days 1–5 followed by a single dose of 3 mg/kg on day 10). Delivery by an elective C-section occurred at 38 weeks of gestation. Upon detection of the Leishmania parasites in the bone marrow, the baby was treated with sodium stibogluconate at an intravenous dose of 20 mg/kg for 20 days and discharged after 30 days [52]. The third one was detected in a baby born to a mother with HIV co-infection immediately after vaginal birth [53]. Leishmania amastigotes were found during the microscopic examination of the placenta, and diagnosis was confirmed by a polymerase chain reaction. The baby was administered intravenous L-AmB at dosage of 5 mg/kg per day from days 1 to 4 followed by a weekly injection of 5 mg/kg and recovered successfully. The mother had received L-AmB 15 weeks after her last menstrual period at a total dose of 40 mg/kg, followed by secondary prophylaxis with a dose of 5 mg/kg every 15 days until delivery.

**Pentavalent antimony (n = 93; 1926–2010).** There were 93 pregnant women who were treated with pentavalent antimony, of whom 70 (75.3%) were from Africa (1984–2009), 19 (20.4%) from Asia (1955–2010), 2 (2.2%) from Europe (1926–1995) and 2 (2.2%) from South America (1994–2003) with majority of the cases treated between 2000–2010 (82 of 93 cases). Treatment was administered to 20 (21.5%) pregnant women in the first trimester, 46 (49.5%) in the second, 22 (23.7%) in the third, while the time in pregnancy was not clear in 5 (5.4%).

Survival status was available for 88 (94.6%) pregnant women of whom 4 (4.5%, 4/88) died due to hepatic encephalopathy [12]. There were 24 (27.3%) abortions (or spontaneous abortions) of which 13 were among African patients and 11 among patients from Asia (Table 3). There were 2 (2.2%) miscarriages, 2 (2.2%) pre-term births, and 1 (1.1%) required splenectomy after delivery due to poor recovery and due to persistence of parasitaemia in the spleen sample (Table 3) [54]. There were 3 foetal deaths reported; one of the babies died due to myelomeningocele 3 hours after birth [12], another died a day after being born [55], and another died due to VL at 2 months [12]. Further details are presented in Table 3.

There were 3 cases of vertical transmission identified [18, 19, 56], detected at 6, 7, and 12 months after birth. The first case (reported in 1926 in the UK) was born to a woman who acquired VL during her third trimester and was treated with urea stibamine [19]. The baby had a normal birth and vertical transmission was detected at 12 months after birth. Upon detection, the baby was treated with antimony and made a recovery. The second case (reported in 1955 in India) was born to a woman in whom the disease was identified during the second trimester and was treated with urea stibamine [56]. Vertical transmission was detected at 6 months; the baby was treated with antimony and was cured after one recurrence. The third case (reported in Sudan in 1922) was born to a woman in whom the disease was identified during the second trimester and was treated with sodium stibogluconate for 30 days [18]. At birth, the baby had signs of intrauterine growth retardation. The baby was diagnosed with VL at 7 months and died two days later (Table 4).

**Amphotericin B deoxycholate (n = 20; 1993–2015).** Of the 20 pregnant women treated with amphotericin B deoxycholate, 17 (85.0%) were from Asia (1993–2015) and 3 (15.0%) from South America (2003–2008). There were 3 (15.0%) pregnant women in their first trimester, 6 (30.0%) in the second, 3 (15.0%) in the third, and the trimester was not clear in 8 (40.0%). There was one (5.0%) maternal death after 7 days of treatment due to haemorrhagic complications occurring after delivery; the pregnant woman was in 28.7 ± 7.8 weeks of pregnancy [57]. The remaining 19 pregnant women were discharged alive. The delivery of babies were described as normal for 18 pregnant women, haemorrhagic complication occurred in one after delivery (as described earlier) [57], and the information was not reported on 1 (Table 2 and S2 Data). There was no reports of vertical transmission of the disease in 12 (60.0%) babies in whom the information was reported (11 were followed-up for 12-months and the status was reported at birth for 1 baby). Further details are presented supplemental file (S2 Data).

**Pentavalent antimony plus paromomycin (aminosidine) (n = 11; 1993–2017).** Eleven pregnant women (all from Africa) were treated with sodium stibogluconate plus aminosidine (paromomycin) [27, 58]. Information regarding trimester, maternal survival status, or vertical transmission were not available (see S2 Data). One spontaneous abortion was reported [58].

**Liposomal amphotericin B plus pentavalent antimony (n = 4; 2006).** Four pregnant women were treated with the combination regimen in a study from Sudan published in 2006 [59], of whom two were in their second trimester and two in their third. L-AmB was administered at 3–7 mg/kg/day on days 1, 6, 11 and 16 (or on days 1, 2, 3, 4, 10 and 15), followed by a 20 mg/kg dose of sodium stibogluconate intramuscularly once daily for 30 days. All four pregnant women were discharged alive. At discharge, one pregnant woman delivered a healthy baby and the remaining three were still pregnant–no follow-up data was reported.

**Paromomycin (aminosidine) (n = 3; 2011).** Three pregnant women were treated with paromomycin in a study from India [60]. The drug regimen was administered at 11 mg/kg/day (base) by deep gluteal intramuscular injection once daily for 21 consecutive days [60]. All three babies were described as being healthy at birth.

**Trimothoprim, sulphadiazine and tinidazole (n = 1; 1994).** One study from India treated a pregnant woman in the first trimester with the combination of Trimothoprim, sulphadiazine and tinidazole [61]. There were no reports of side effects and the pregnant woman delivered a healthy baby at term. The mother remained well at one year after completion of treatment with no further information available on the infant.

**Unclear drug substance (n = 12;1987–2009).** Three publications reported treating pregnant women with VL in whom either the drug name was not specified or the allocated drug group (in multi-arm study) could not be identified. Two publications described a case each without reporting the name of the drug administered [17, 62]. In the third report, the number of pregnant women (n = 10) allocated to each drug arm (pentavalent antimony or amphotericin B deoxycholate) was not clear [63]. Two (16.7%) of the pregnant women were in their third trimester and the status was unknown for the remaining 10 (83.3%). There were two (16.7%) maternal deaths [63] and two cases of vertical transmission [17, 62]. The first one was identified at 8 months after birth and another at 6 weeks after birth (the baby died after 3 days). Both babies were administered treatment upon detection of VL (See S2 Data for further details).

**Untreated (n = 3; 2001–2017).** Three cases of VL identified during pregnancy were untreated [25, 64, 65]. Treatment was deferred until after delivery due to safety concerns in one report in a pregnant women in third trimester [25]; there were signs of intra-uterine growth retardation requiring an emergency C-section, and both mother and the child were alive. The second pregnant woman in third trimester was not treated due to lack of adequate hospital resources [64]; the baby born to the mother died after 2 months due to malnutrition with no evidence of vertical transmission. In the third case, VL was diagnosed nearer the delivery time before treatment could be administered but the mother died after giving birth and before treatment could be administered [65]; the baby also died and foetal part placenta examination revealed presence of Leishman Donovan bodies by polymerase chain reaction indicating vertical transmission.

## Confirmed/probable/suspected vertical transmission

We identified a total of 26 cases of confirmed, probable or suspected cases of vertical transmission (Table 4). The median time to detect vertically transmitted VL was 6 months (range: 0–18 months). Eleven babies were born to women in whom the disease status was confirmed during their pregnancy; three of the mothers were treated with L-AmB, three with PA, the drug name was not clear in two, a mother was untreated and the treatment status was not clear in the remaining two (Table 4). Histopathological examination of the placenta confirmed the vertical transmission of the disease in two cases [18, 65] and this was not reported for the remaining cases. Treatment status was described in 18 children, of whom 11 received pentavalent antimony, 6 received L-AmB and 1 received amphotericin b deoxycholate. Two of the children died; one had received pentavalent antimony and the drug name was not clear in the other (See S2 Data for further details).

## Risk of bias assessment

Of the 71 case reports/case series, 49 (69.0%) were judged to be at a high risk of bias for patient selection and the remaining 22 (31.0%) were at low risk of bias. For the ascertainment of exposure status, 6 (8.5%) were at high risk of bias, 17 (23.9%) were at moderate risk, 41 (57.7%) at

low risk, and the risk of bias status was not clear in 7 (9.9%) studies. For the adequacy of follow-up, four (5.6%) studies were considered to be at moderate-high risk of bias, 37 (52.1%) at low risk of bias, 23 (32.4%) were retrospective studies, and the risk of bias was unclear in 7 (9.9%). For the completeness of reporting domain, 5 (7.0%) studies were at high risk of incomplete reporting, 20 (28.2%) at moderate risk of bias, and 46 (64.8%) were at low risk of bias (See S1 Table). One retrospective cohort study with a comparative group of non-pregnant patient group was considered of high quality.

## Discussion

The occurrence and effects of VL during pregnancy is under-researched and poorly understood as evidenced by having identified only 72 publications describing a total of 451 cases of VL in pregnancy in the past 94 years.

The small case volume reported in the literature could have several explanations. In the first place, there is an apparent imbalance in caseloads with predominance of the disease among males; ascribed to biological or behavioural causes [10, 13, 73, 114–116]. Pentavalent antimony was the first line therapy before the development of Liposomal amphotericin B (L-AmB)–this might have traditionally dissuaded physicians from treating VL during pregnancy and leaning towards postponing the treatment until after delivery unless treatment is absolutely warranted [25]. However, this situation might have changed recently as L-AmB has no contraindication during pregnancy and is the treatment of choice adopted in national guidelines (Table 1). It has also been postulated that early pregnancies are missed due to spontaneous abortion caused by VL [73]. Women with childbearing potential or those who are already pregnant are systematically excluded from VL clinical studies and only a third of the patients enrolled in clinical trials are females [10]. It is clear that the likely caseload of VL in pregnancy is much bigger than what can be estimated from available reports. For example, during Jan 2016–Jul 2019 in Lankien, Jonglei state, South Sudan, out of 4,448 cases of VL diagnosed, 824 (18.5%) occurred in females aged 15–44 years and 110 (13%, 110/824) of them were pregnant (2.5% of all cases) (Personal communication with Dr. Ritmeijer).

There was also geographical disparity in the treatment regimens used. Among the cases identified during pregnancy, just over two-thirds of the patients received amphotericin B regimens (any formulations or in combination with another drug) in studies conducted in Africa compared to more than 75% of the patients from Asia. This could reflect either heterogeneity in treatment practices or the differential parasite susceptibility against drug regimens in the regions.

Among 346 cases in whom VL was diagnosed during pregnancy and received a treatment, there were a total of 11 (3.2%) maternal deaths reported; four (4.5%, 4/88) occurred in those treated with pentavalent antimony-based regimens, 4 (4/176; 2.3%) among those treated with L-AmB, 1 (1/20, 5.0%) with amphotericin b deoxycholate, and the drug name used for the treatment was not clear in 2 (16.7%,2/12) cases. Abortion following PA regimen was observed in just over a quarter of the pregnant women (24/88, 27.3%) of whom 13 were from Africa and 11 were from Indian sub-continent. Following the L-AmB regimen, there were a total of 5 (2.8%) miscarriages and 1 (0.6%) foetal death and no cases of abortion were reported (Table 3). Taken together, these results support the use of liposomal amphotericin B for the treatment of VL during pregnancy.

Our review identified 26 confirmed/suspected/possible cases of vertically transmitted VL with a median time of detection of 6 months (range: 0–18 months); 12 (46%) cases were identified after 6 months of birth. This suggests that children born to mothers with VL during pregnancy may require a longer post-treatment follow-up than the standard 6-months follow-up

duration among non-pregnant patients to identify cases of vertical transmission. The underlying mechanism of the onset of clinical leishmaniasis among neonates and infants born to a successfully-treated mother during pregnancy is currently not clear; it has been ascribed to imbalances in immune-mechanism modulated by T cell responses (Th1/Th2) [24] or by parasites entering a state of dormancy in the lymph nodes [81].

Our review has identified limitations in reporting of VL in pregnancy. Complete information was often not available on treatment administered or on efficacy and safety outcomes for the mother and baby. For 12% (54/451) of the pregnant women, it could not be ascertained whether they had received any treatment or not. Just over two-third of the studies were considered to be a high risk of bias for patient selection while a nearly a third of the studies were at moderate to high risk of bias for ascertainment of exposure domain as VL diagnosis was purely based on clinical signs and symptoms or on serological methods (S1 Table). Overall, the large number of case reports/case series included in this review provides an explanation for such high risk of bias and likely indicates that the reported evidence would be regarded as low quality. This review also suggested that existing practices for management of VL in pregnancy is guided by limited evidence generated from case reports and small case series (71 of the 72 studies were case-reports/case series). High quality studies (such as Pekelharing-2020 [11]) is warranted for generation of a robust evidence regarding safety and efficacy of antileishmanial agents during pregnancy. There was also a lack of standardised reporting as information was missing on several critical parameters such as trimester status, time of detection of VL, and therapeutic outcomes for the mother and the child. Taken together, these findings highlight the need to improve and harmonise the reporting of VL in pregnant women. We have outlined a minimum checklist of items that might be useful for reporting purposes (Box 1).

| Domain | Checklist Item |
|---|---|
| Maternal history | Parity |
| | Gravidity |
| | Maternal history of the disease |
| | History of travelling to endemic regions |
| | Any previous treatment of the disease |
| | Comorbidities (HIV, malaria, TB etc.) |
| Maternal characteristics | Age |
| | Weight |
| | Nutritional status |
| | Trimester |
| | Gestational age<br>Haemoglobin on admission |
| Maternal clinical signs and symptoms | History of illness (duration of fever) |
| | Hepatomegaly status |
| | Splenomegaly |
| Diagnostics | Diagnostic method used (PCR, ELISA, rk39DAT, IFA) |
| | Sample analysed (blood, bone marrow aspirates, splenic aspirates etc) |
| | Method of confirmation of disease status |
| | Parasite species (*L. donovani*, *L. infantum*) |
| | Method used for estimating gestational age (ultrasonography etc) |
| Treatment details | Dose, duration, frequency including mode of administration |

(*Continued*)

**Table 4.** (Continued)

|  |  |
|---|---|
|  | Concomitant medication status (antipyretics, antimalarial etc) |
|  | Need for blood transfusion before or during treatment |
| Delivery characteristics | Mode of delivery (C-section, natural) |
|  | Trauma during delivery |
|  | APGAR score |
|  | Examination of placenta |
|  | Birth status (still birth, abortion, healthy birth) |
|  | Any birth-related complications |
| Outcomes | Clinical response (initial cure/default/non-response/death) |
|  | Adverse events |
|  | Fetal outcome status: abortion/stillbirth/premature birth/healthy birth |
|  | Follow-up data on babies |

PCR = Polymerase chain reaction; ELISA = enzyme-linked immunosorbent assay; IFA = Immunofluorescence assay

In addition to the articles presented in this review, data from women who become pregnant after completion of therapy but within the follow-up period enrolled in trials might provide further evidence on drug safety, especially on the reproductive consequences of the treatment (Table 5). The recently proposed safe ethical framework for the recruitment of women susceptible to and becoming pregnant is an important development towards filling the existing knowledge gap [28]. Like for many neglected tropical diseases, there is currently an absence of a comprehensive pregnancy-specific registry for exposures to antileishmanials, with the exception of the one dedicated for miltefosine [117]. Therefore, creating an open registry where all these cases are indexed and continually updated would help in better characterisation of the safety aspects of the drugs. Finally, the Infectious Diseases Data Observatory (IDDO) data platform, that is currently standardising individual participant data from several VL clinical studies, offers a unique resource to explore host, parasite, and drug dynamics affecting the safety and efficacy in pregnant populations [118].

**Table 5.  Description of patients enrolled in clinical trials who became pregnant after completion of treatment.**

| Study | Number of patients | Treatment received at enrolment | Pregnancy and outcome description |
|---|---|---|---|
| Bhattacharya-2007 [119] | 2 | Miltefosine | "Despite extensive counselling for contraception, 2 cases of pregnancy were reported, with the conception date close to the exposure period. One patient became pregnant 2 weeks after the end of treatment, and the other became pregnant at 3 months after the end of the treatment period. Two healthy babies were delivered at gestational weeks of 39 and 40, without any birth anomaly" |
| Sinha-2011 [60] | 1 | Paromomycin | One female patient became pregnant more than 1 month after completing treatment. The offspring was born alive and determined to be normal/healthy just after birth. |
| Mondal-2014 [120] | 4 | Liposomal amphotericin B (single dose) | Four female participants became pregnant within months after treatment. In one, the pregnancy was completed with delivery of a term/normal birth after 6 months of follow-up. The other three pregnant women were clinically healthy during the last follow-up visit. |
| Jamil-2015 [121] | 1 | Paromomycin | Pregnancy was reported in one female during the follow-up period. The offspring was born healthy and a hearing test conducted on the infant at 1.5 months of age confirmed reaction to sound. An otoscopy and oto-acoustic emission test to determine function of the middle and inner ear was conducted at 3 months of age and confirmed normal hearing function. |
| Pandey-2016 [122] | 15 | Miltefosine | Fifteen patients became pregnant within 6 months of follow-up (all these patients became pregnant 2 months after end of the treatment). All of them were followed for one year and all had full-term normal pregnancy with no congenital anomalies. |

## Conclusions

In conclusion, this review brings together scattered observations on VL in pregnant women and the cases of vertically transmitted VL reported in the clinical literature. Available reports clearly underestimate the scale of the problem. Existing therapeutic guidelines regarding the usage of drugs in pregnancy is guided by limited evidence generated from case reports and small case series. Our review suggests that liposomal amphotericin B should be the preferred treatment for VL during pregnancy.

## Supporting information

**S1 Text. PRISMA checklist.**
(DOCX)

**S2 Text. Search details.**
(DOCX)

**S1 Data. Screening list.**
(XLSX)

**S2 Data. Study data.**
(XLSX)

**S1 Table. Risk of bias assessment.**
(XLSX)

## Acknowledgments

We would like to thank for the Prof. Bernhard Lämmle and his team for helpful responses on queries related to their manuscript. We would like to thank Christina Woodward for locating some of the articles included in the review.

## Author Contributions

**Conceptualization:** Prabin Dahal, Philippe J. Guerin, Piero L. Olliaro.

**Data curation:** Prabin Dahal, Sauman Singh-Phulgenda.

**Formal analysis:** Prabin Dahal, Sauman Singh-Phulgenda, Philippe J. Guerin, Piero L. Olliaro.

**Funding acquisition:** Philippe J. Guerin.

**Investigation:** Prabin Dahal, Sauman Singh-Phulgenda, Philippe J. Guerin, Piero L. Olliaro.

**Methodology:** Prabin Dahal, Sauman Singh-Phulgenda, Brittany J. Maguire, Eli Harriss.

**Project administration:** Prabin Dahal, Piero L. Olliaro.

**Resources:** Philippe J. Guerin, Piero L. Olliaro.

**Software:** Prabin Dahal.

**Supervision:** Philippe J. Guerin, Piero L. Olliaro.

**Validation:** Prabin Dahal, Sauman Singh-Phulgenda, Philippe J. Guerin, Piero L. Olliaro.

**Visualization:** Prabin Dahal.

**Writing – original draft:** Prabin Dahal, Piero L. Olliaro.

**Writing – review & editing:** Prabin Dahal, Sauman Singh-Phulgenda, Brittany J. Maguire, Eli Harriss, Koert Ritmeijer, Fabiana Alves, Philippe J. Guerin, Piero L. Olliaro.

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
