## [Decision Letter · Decision Letter 0]

3 May 2021

Dear Dr Dahal,

Thank you very much for submitting your manuscript "Visceral Leishmaniasis in pregnancy and vertical transmission: A systematic literature review on the therapeutic orphans" for consideration at PLOS Neglected Tropical Diseases. As with all papers reviewed by the journal, your manuscript was reviewed by members of the editorial board and by several independent reviewers. The reviewers appreciated the attention to an important topic. Based on the reviews, we are likely to accept this manuscript for publication, providing that you modify the manuscript according to the review recommendations. 

Sincerely,

Walderez O. Dutra, PhD.

Deputy Editor

Walderez Dutra

Deputy Editor

Reviewer's Responses to Questions

**Key Review Criteria Required for Acceptance?**

**Methods**

-Are the objectives of the study clearly articulated with a clear testable hypothesis stated?

-Is the study design appropriate to address the stated objectives?

-Is the population clearly described and appropriate for the hypothesis being tested?

-Is the sample size sufficient to ensure adequate power to address the hypothesis being tested?

-Were correct statistical analysis used to support conclusions?

-Are there concerns about ethical or regulatory requirements being met?

Reviewer #1: Yes

Reviewer #2: The review is appropriate to address the authors' questions and objectives. Statistical analysis and hypothesis testing are not relevant. There are no ethical concerns.

Reviewer #3: - Objectives & Methods are clearly stated. 

- Given the heterogeneity and nature of studies (cases series, no RCTs) comparative statistics were not appropriate. 

This was clearly explained. 

Descriptive statitistics are sufficient and clear. 

- PRISMA flow diagram & Supplementary material: very clear

**Results**

-Does the analysis presented match the analysis plan?

-Are the results clearly and completely presented?

-Are the figures (Tables, Images) of sufficient quality for clarity?

Reviewer #1: Yes

Reviewer #2: The analysis, results, and corresponding tables are complete and sufficiently clear.

Reviewer #3: - Yes

- Results clearly presented.

- Tables & figures are very clear.

**Conclusions**

-Are the conclusions supported by the data presented?

-Are the limitations of analysis clearly described?

-Do the authors discuss how these data can be helpful to advance our understanding of the topic under study?

-Is public health relevance addressed?

Reviewer #1: Yes

Reviewer #2: Conclusions are supported by the data presented. The authors describe some limitations, acknowledge relevance to physicians, and further our understanding of VL treatment among an understudied yet high-risk group.

Reviewer #3: - Yes - large body of evidence with > 200 studies and > 400 pregnant women included. 

- Risk of bias assessment is clear. Transparent on heteregeneity and scarcity (1) of comparative cohorts. 

- Yes - usefulness well discussed

- Yes - public health well addressed. 

(Large number of studies from a wide geographical area).

**Editorial and Data Presentation Modifications?**

Reviewer #1: Review comments

Line 93-95: 

This sentence is confusing. The ‘in utero’ congenital transmission is considered trans-placental. What is the supporting evidence for the congenital transmission through blood exchange during labour? 

According to Berger, 2017 (ref no. 10): “To date, the transplacental route is the only route that has been described for CT of Leishmania, and these studies represent ample proof of the capacity of Leishmania for placental invasion and CT via the transplacental route, based on demonstration of parasites in placental, fetal, and newborn tissue.”

Line 97-98:

Such vertical transmission can induce in utero death or can be potentially deleterious to the foetus and infant. 

Consider moving this sentence before the previous sentence. 

Now you talk about reports for vertical transmission with a manifestation several months postpartum, and in the next sentence ‘such vertical transmission can cause intra uterine death’. 

Line 117:

Typo Expect = except 

Line 127-132:

What is expected to be the added value of this literature review? 

In the previous 2 reviews that are mentioned (Pagliano and Figuero, ref 9,22) it was already concluded that treatment of VL in pregnancy is imperative and that liposomal Amphotericin B is the drug of choice. 

Also in the WHO report on Control of the Leishmaniasis from 2010 , liposomal Amphotericin B was already mentioned as the drug of choice for VL in pregnancy. 

However this review is more comprehensive and includes recent publications which makes it a valuable update of knowledge on VL in pregnancy.

Line 210:

Typo Were 69 were

Line 260: 

Ref 43 – splenectomy after birth? What was the indication for splenectomy? Suspected bleeding? Splenectomy is not a normal treatment intervention for VL. 

Line 261:

The baby who died at 2 months age due to VL, is this not considered vertical transmission? This baby is not mentioned in line 262-263. 

Line 360-366:

Consider removing the calculation as it is probably far off? 

Line 373:

It would be informative to mention the (relative) disease burden per country. Some years back South Sudan had the second highest number of VL cases in the world after India, so it is not only a matter of local interest. Which countries have a high disease burden which is not represented in the available data? 

373-384:

There were different treatment regimens. What are the causes for this? The time frame of the included studies is very wide, so treatment regimens have changed over time. Increased availability of liposomal Amphotericin B over time must have played a role. As it is costly and it needs cold chain it may not be available in all resource-poor settings. What is considered the second best alternative in case Liposomal Amphotericin B is not available? 

What is the role of the different strains of parasites in the different continents? The susceptibility to Amphotericin B is better in Asia than in Africa, I believe. 

I miss interpretation of the outcomes in the context of time and setting. 

(https://www.ncbi.nlm.nih.gov/pmc/articles/PMC2889656/

In zoonotic VL (the Mediterranean Basin, the Middle East, and Brazil) a total liposomal amphotericin B dose of ≥20 mg/kg is adequate to treat immunocompetent children and adults in these regions. The exact dosing schedule can be flexible (divided into doses of 10 mg/kg on 2 consecutive days or in smaller divided doses), but liposomal amphotericin B pharmacokinetics suggest that the initial dose will provide better tissue levels if at least 5 mg/kg is given. The schedule of 10 mg/kg/day on two consecutive days needs to be validated in adults.

For HIV-VL coinfection, Highly Active Antiretroviral Therapy should be a priority. There is an urgent need for multicenter trials of L-AmB as a first-line treatment and for secondary prophylaxis of VL in HIV-infected patients. In the anthroponotic cycle in the Horn of Africa, liposomal amphotericin B can be given at a total dose of 20 mg/kg however, a dose of 10–15 mg/kg may be adequate for South Asia.

In India, where a single infusion of L-AmB by itself at 7.5 or 5 mg/kg can induce cure rates of 90%–91%, combination regimen with lower doses of L-AmB with miltefosine or paromomycin is an option. With the preferential pricing, along with just one day of hospitalization, makes a single infusion of 10 mg/kg of L-AmB considerably less expensive and a viable option for the treatment of VL in the subcontinent.[53]. Well conducted trials of combination therapy with L-AmB is urgently needed.)

Line 396:

You identified a high risk of bias. What could this mean for the presented data? In other words do you expect over- or underestimation of cure rates/ mortality, what does it mean for the follow up data? 

Did you find a risk factor for vertical transmission? E.g. in which trimester the mother was affected, duration / type of treatment. 

For the checklist (Box 1) consider adding: 

Hb on admission, 

Severity score (https://journals.plos.org/plosntds/article?id=10.1371/journal.pntd.0005921), 

Need for blood transfusion during treatment, 

Treatment response (clinical cure, parasitological cure) 

Total dose of treatment given 

Side effects 

(Initial) Treatment outcome: Initial cure / non-response / defaulter / death

Ideally: Definite cure (at 6 months)

Delivery during treatment ? (abortion / stillbirth / premature birth / healthy birth)

Delivery complications (PPH?)

Ideally: Pregnancy outcome after completing treatment (abortion / stillbirth / premature birth / healthy birth)

Ideally: follow up of neonate born to mother treated during pregnancy (survival, signs of vertical transmission)

Reviewer #2: • In lines 105-106, delays in treatment of PKDL do not appear to be relevant. 

• For clarity, I recommend removing “(for example in miltefosine trials)” in line 112. Miltefosine is not formally introduced in the text until later, and the South Asian study referenced in line 115 isn’t a miltefosine trial. Consider including something along the lines of, “A study conducted in South Asia found that only one in every six doctors ruled out pregnancy before prescribing miltefosine, a medication which has been contraindicated in pregnancy.” 

• It is unclear how Table 1 is organized. I recommend using alphabetical order for simplicity.

• The meaning of lines 234-236 is unclear: “Ten were suspected of having carried the infection during their pregnancy, of whom 6 were cases of sub-clinical persistence of the parasites without the mother ever suffering from the disease previously.” Does this mean the parasites were present in women who were recently pregnant? 

• I recommend clarifying that outcomes are presented by treatment type in line 240.

• Although a bit redundant, readers would greatly benefit from transitions at the beginning of each treatment paragraph. It would be helpful if line 243 began with: “Among mothers treated with Liposomal Amphotericin (n=202)…”. Similarly, it would be helpful if line 255 began with: “Among mothers treated with pentavalent antimony (n=92)…” The authors do a good job providing context in lines 268 (amphotericin B deoxycholate), 279 (aminosidine), 290 (paromomycin), 294 (unclear), and 302 (untreated). 

• For clarity, I recommend rephrasing lines 247-248. Consider something along the lines of, “There were five (2.9%) miscarriages, 1 (0.6%) stillbirth, and 1 (0.6%) premature birth. The mother’s trimester at the time of these outcomes was unclear. One fetal death was reported from a mother in her first trimester.”

• In line 271, how was it determined that the mother was about 29 weeks pregnant? Was this the duration range of the mothers included in the referenced study? 

• In line 361, please spell out IDDO before the acronym.

• In lines 387-388, it is unclear why 6 months follow-up after treatment is insufficient.

Reviewer #3: Minor modifications: 

- ABSTRACT/Results part: 

Outcomes were reported in 176 mothers treated (add "by") L-AmB with 4 (2.3%) reports of maternal deaths,... 

- Line 116-117: "Finally, there is a lack of active pregnancy registries for most of the antileishmanial drugs

117 expect for miltefosine."  change expect to EXCEPT for... 

- Line 117: Impavo® (Profounda Inc.)  change to ImpaVIDO

**Summary and General Comments**

Reviewer #1: A comprehensive review on an understudied topic. It adds to the body of evidence for treatment of VL in pregnancy. I appreciate the suggestion in Box 1 "Proposed minimum variable recording and reporting for studies or case reports for VL in pregnancy" and also the comprehensive description of the reported cases on vertical transmission. 

In the discussion I miss interpretation of the results in the context of time and setting.

Reviewer #2: I appreciate the work of the authors on reviewing the literature about reported cases of VL in pregnant women. The determination of the safest VL treatments for mother-infant pairs, and the examination of cases of VL vertical transmission are valuable to physicians who typically relied on thin precedence to administer treatment. The synthesis of information about the treatment of VL in a vulnerable yet understudied population is meaningful. The comprehensive capture of cases from multiple literature sources and across 90 years is a strength of this review. However, I have highlighted below some concerns about insufficient context that should be addressed in the paper:

MAJOR COMMENTS

Introduction:

• I believe the authors will strengthen this review by adding a paragraph at the beginning of the introduction describing the signs, symptoms, and severity of VL. Is the disease short-term or long-term? Is it fatal? A quick search indicates a wide clinical spectrum including fever, weight loss, and severe anemia. This information is helpful to include because it provides more comprehensive context about VL’s burden. 

• I recommend adding a paragraph between the first and second paragraphs of the introduction (lines 85 and 86) about the typical approach to diagnosing and treating VL among non-pregnant women. The first lines of paragraph two indicate that diagnosis among pregnant women relies on symptoms and serology. Does diagnosis not typically involve symptoms and serology? Is splenic aspiration the typical approach to diagnosing VL? Line 88 mentions more severe anemia and increased requirements for blood transfusion among pregnant women. Is VL associated with anemia in the wider population? Does VL exacerbate existing anemia or is it associated with new-onset anemia? Are blood transfusions common among non-pregnant VL patients? Describing how treatment during pregnancy deviates from the norm brings to the foreground the particular risks of VL to mothers and infants. 

Methods:

• I appreciate the inclusion of several types of published literature formats to capture as many cases as possible! 

• Assessment of bias was helpful, but the authors would benefit from describing the classification of risk of bias for outcome assessment, adequacy of follow-up, and reporting of results. The authors mention that the risk of selection bias was high for single case reports and lower for cohort studies. This was very helpful. For outcome assessment, how did the authors ascertain low or high risk of bias? What duration of follow-up was classified as higher or lower risk bias? What made how a case was reported lead to a classification of higher/lower risk of bias? I recommend describing these parameters in the same way selection bias and ascertainment of exposure were described. Expand on the findings in the results section.

Results:

• The authors did a good job of describing outcomes among the mothers and infants. However, the authors tended to mention important information such as dosing, treatment information, and latter outcomes briefly in parentheses instead of describing the cases in greater detail. For example, in line 252, the authors write, “… and for the third case, vertical transmission was suspected 8 months after birth (treated with sodium stibogluconate 20 mg/kg for 20 days and discharged)”. This is one of only 3 cases of vertical transmission among those treated with L-AmB, so its circumstances merit commentary. A glance at the abstract indicates that the mother was co-infected with HIV. This is an extenuating circumstance that might complicate simple vertical transmission. Consider eliminating parentheses around important information, especially when describing specific or rare instances. See also:

o Lines 250, 264-265, 271-272, 315-317, and 321

• I recommend describing the time trends and geographic locations of the most common adverse birth outcomes to build a case for the dangers of a particular treatment. For example, 27% of women treated with pentavalent antimony experienced abortion or spontaneous abortion (line 258). Did all of these women live in the same country/continent? Were these abortions consistent over time, or were they more common a few decades ago? Did each woman receive the same (or similar) dose? It is clear that pentavalent antimony is associated with worse outcomes, but the context of these outcomes strengthens the evidence that the drug itself, and not alternative circumstances like delayed treatment, severe VL, or a lack of access to care, are to blame.

• The results paragraph on untreated women (line 301) is the strongest in my opinion because it includes context about the extenuating circumstances of the reported adverse outcomes. I appreciate that the authors point out safety concerns, the rationale for a c-section, a lack of hospital resources, and suspected vertical transmission among the untreated women in this section. When possible, I recommend the authors include similar details about extenuating circumstances in other areas of the results section. Very interesting to read!

Discussion

• I believe the authors would strengthen the discussion by expanding on the possibility of time trends in treatment regimens. Given that studies of VL cases spanned 90 years, what changes have been observed over time? The authors mention that prevalent antimony has been contraindicated in pregnancy (line 352). Is it known when this occurred? When did L-AmB become the “treatment of choice” (line 357)? The authors also mention that more than half of cases occurred after 2005 (line 372). Has the standard course of treatment changed in the past 16 years?

• The line of thinking in lines 360-370 is very confusing. It’s not clear where the 15,883 included patients come from (line 363) or how the authors estimated 96 cases per 100,000 people. Also, it may not be likely that everyone screened for eligibility had the disease, meaning 96 cases is likely an overestimate. The point taken is that cases of VL among pregnant women are underreported and recorded incidence is probably an underestimate. Since the example from IDDO records does not add to this conclusion, I recommend removing this paragraph for clarity.

• I strongly recommend adding a paragraph or two about geographic trends. The authors mention as an aside in line 371 that more than half of cases came from 2 countries. I believe that is a significant point that merits additional commentary. How might overrepresentation from Sudan and South Sudan impact results and conclusions? Is a certain treatment approach more common in a given region?

• The inclusion of the checklist is a great idea!

• In line 393, I recommend clarifying that the limitations in reporting of VL in pregnancy correlate with limitations in the present work. The review is comprehensive but limited because it makes use of cases aside from peer-reviewed studies and included some studies translated through Google Translate. Conclusions are based on the over-representation of cases in Sudan and South Sudan, which may differ from treatment approaches in other nations.

Reviewer #3: Very useful systematic review.

Under-researched topic.

Unique body of evidence. 

Large number of studies from a wide geographical area. 

Your final sentence from the "Authors summary" is very relevant: our findings indicate that L-AmB should be the preferred treatment for VL

79 during pregnancy. " 

This could sentence could also fit in the abstract, where it appears less clearly (more cautiously): maybe too cautiously given the big difference in mortality% and miscarriage%. 

This adds some evidence on the danger of using antimonials during pregnancy. (despite the studies not being comparative and from different time periods).

PLOS authors have the option to publish the peer review history of their article (what does this mean?). If published, this will include your full peer review and any attached files.

Reviewer #1: No

Reviewer #2: No

Reviewer #3: Yes: Gabriel Alcoba

Figure Files:

Data Requirements:

Reproducibility:

References

---

## [Editor Report · Decision Letter 1]

13 Jul 2021

Dear Dr Dahal,

We are pleased to inform you that your manuscript 'Visceral Leishmaniasis in pregnancy and vertical transmission: A systematic literature review on the therapeutic orphans' has been provisionally accepted for publication in PLOS Neglected Tropical Diseases.

Best regards,

Walderez O. Dutra, PhD.

Deputy Editor

Walderez Dutra

Deputy Editor

---

## [Editor Report · Acceptance letter]

5 Aug 2021

Dear Dr Dahal,

We are delighted to inform you that your manuscript, "Visceral Leishmaniasis in pregnancy and vertical transmission: A systematic literature review on the therapeutic orphans," has been formally accepted for publication in PLOS Neglected Tropical Diseases.

Best regards,

Shaden Kamhawi

co-Editor-in-Chief

Paul Brindley

co-Editor-in-Chief
